# Compressed Graphene Assembled Film with Tunable Electrical Conductivity

**DOI:** 10.3390/ma16020526

**Published:** 2023-01-05

**Authors:** Qiang Chen, Zhe Wang, Huihui Jin, Xin Zhao, Hao Feng, Peng Li, Daping He

**Affiliations:** 1Hubei Engineering Research Center of RF-Microwave Technology and Application, Wuhan University of Technology, Wuhan 430070, China; 2State Key Laboratory of Advanced Technology for Materials Synthesis and Processing, Wuhan University of Technology, Wuhan 430070, China; 3School of Information Engineering, Wuhan University of Technology, Wuhan 430070, China

**Keywords:** graphene assembled film, high electrical conductivity, fabrication process, strain sensor

## Abstract

Graphene and graphene-based materials gifted with high electrical conductivity are potential alternatives in various related fields. However, the electrical conductivity of the macro-graphene materials is much lower than their metal counterparts. Herein, we improved the electrical conductivity of reduced graphene oxide (rGO) based graphene assembled films (GAFs) by applying a series of compressive stress and systematically investigated the relationship between the compressive stress and the electrical conductivity. The result indicates that with increasing applied compressive stress, the sheet resistance increased as well, while the thickness decreased. Under the combined effect of these two competing factors, the number of charge carriers per unit volume increased dramatically, and the conductivity of compressed GAFs (c-GAFs) showed an initial increasing trend as we applied higher pressure and reached a maximum of 5.37 × 10^5^ S/m at the optimal stress of 450 MPa with a subsequent decrease with stress at 550 MPa. Furthermore, the c-GAFs were fabricated into strain sensors and showed better stability and sensitivity compared with GAF-based sensors. This work revealed the mechanism of the tunable conductivity and presented a facile and universal method for improving the electrical conductivity of macro-graphene materials in a controllable manner and proved the potential applications of such materials in flexible electronics like antennas, sensors, and wearable devices.

## 1. Introduction

Graphene, as the thinnest two-dimensional material with a honeycomb lattice nanostructure that contains numerous double bonds, has been attracting prevalent attention [1,2] in the fields of electronics [3,4], thermodynamics [5,6,7,8], mechanics [9,10,11], optics [12,13,14] and chemistry [15,16,17]. Many experimental studies have been carried out focusing on synthesizing macro graphene and graphene-based materials via various methods to improve the electrical and mechanical properties for potential applications [18,19,20,21,22,23,24]. For instance, Zhou et al. obtained Mxene-functionalized rGO with high compactness and toughness through Ti-O-C covalent bonding to improve the poor mechanical properties of graphene [25]. Wan et al. succeeded in the scalable fabrication of graphene sheets with high strength of 1.55 GPa and improved electrical conductivity by freezing stretch-induced alignment for graphene layers [26]. Different reduction methods for graphene oxide (GO) to tune the negative Poisson’s ratio for auxetic reduced GO film were also reported by researchers [27,28]. By growing the graphene on metal surfaces, Deck et al. reported an excellent exciton-dominated optical response [29]. Wu et al. achieved the tuning of thermal conductance for graphene composites by filling the graphene layer [30]. These works inspired further research and contributed to genuine progress in this field.

Electrical conductivity is an extremely important fundamental property that represents the ability of the material to conduct current. It is also one of the determining properties of graphene and graphene-based materials for electrical applications such as sensors, electromagnetic shielding, antennas, and flexible electronics. Since the discovery of graphene, various efforts have been made to improve its electrical conductivity. For example, the relation between the distances among neighbor holes and the conductivity of graphene was predicted by simulation by Pavel et al. [31]. To improve the electrical conductivity, stable, effective p-doped graphene with tunable graphene electronics has been obtained under local ultrahigh pressure using atomic force microscopy (AFM) diamond tips by Pablo et al. [32]. Valcheva et al. fabricated the freestanding graphene sheets by microwave plasma at atmospheric pressure with electrical conductivity of about 10^3^ S/m [33]. However, none of these can compete with their metal counterpart Cu, with electrical conductivity of 5.7 × 10^7^ S/m.

Given the widest applicability of film materials within common macro graphene materials, this work investigated subsequent compression for improving the electrical conductivity based on graphene-assembled films (GAF) by thermal annealing. The result showed that the electrical conductivity of GAF could be regulated by applied pressure and peaked at 5.37 × 10^5^ S/m under the optimal compressive stress of 450 MPa and then decreased. Further, the obtained compressed graphene films were applied as strain sensors which showed good stability and sensitivity. This work provides general guidance on controlling the electrical conductivity for macro graphene materials to achieve different applications [34,35,36,37].

## 2. Materials and Methods

### 2.1. Preparation of GAF

The GO powder purchased from Wuxi Chengyi Education & Technology Co., Ltd. (Wuxi, China) was dispersed in deionized water with solid contents of 2–4%. The obtained suspension was then stirred at 450 rpm for 4–6 h until it transformed into a gel. After carefully pouring the polyethylene terephthalate (PET) substrate into a glass mold at the proper height, the GO gel was dried up at room temperature until the GO film was formed. This newly formed GO film was peeled off from the PET substrate and then thermally annealed with the heating rate of 10 °C/min under argon shielding at 1300 °C for 2 h and 3000 °C for 1 h. The GAF was obtained after cooling to room temperature.

### 2.2. Preparation of c-GAF

The obtained GAF was cut into square or rectangular pieces and was then set between two PET films forming a sandwich-like structure (PET-GAF-PET) to avoid damage from the tiny machine marks on the metal plate on the test machine. The laminated structure was then compressed by a testing machine (MTS with force capacities from 1 to 3000 kN) at a speed of 0.2 mm/min until reaching the set pressure. The pressure was kept constant for 12 h. After the compression process, the PET films were peeled off from the GAF, and the c-GAF was obtained.

### 2.3. Characterizations

The high-resolution transmission electron microscope (HRTEM, JEOL JEM-1400 Plus, Tokyo, Japan) was used to observe the morphology of GO and rGO nanosheets. The cross-sectional images of GO and GAF were captured by the field-emission scanning electron microscope (SEM, Zeiss Ultra Plus, Oberkochen, Germany). The 3D reconstruction for surface morphology and cross-section of the c-GAF specimens were processed by the digital microscope (KEYENCE, VHX-600E, Osaka, Japan). The sheet resistance of c-GAF was measured by the four-point probe system (RTS-9). The defects of molecular structures in GO and GAF were examined by the confocal Raman microscope (HORIBA LabRAM HR Evolution, Kyoto, Japan).

### 2.4. Tensile Test for c-GAF Strain Sensors

The c-GAFs were cut into rectangles (5 mm × 70 mm) and fabricated into strain sensors. The c-GAF strain sensors were fixed on the glass fiber-reinforced plastic (GFRP) by polydimethylsiloxane (PDMS). Foil gages (BE120-5AA) were fixed on the opposite sides of each GFRP specimen to indicate the deformations. All GFRP specimens with strain sensors were stretched by the universal testing machine (Instron 5882, Norwood, MA, USA) at a speed of 0.5 mm/min. The resistance of the c-GAF sensors was measured by a digital multimeter (Keithley 6510, Cleveland, OH, USA). The outputs of the foil gages were recorded by the strain gauge (uTekL uT7110Y, Wuhan, China).

## 3. Result and Discussion

Figure 1a shows the Schematic illustration of the reduction process for GAF. After the evaporation and drying process of the GO solution, the GO film was obtained, which was formed by GO nanosheets with several kinds of oxygen-containing functional groups. During the thermal annealing, the remaining water was first converted into steam. Then the oxygen-containing functional groups were removed from GO nanosheets as gaseous carbon oxides like CO and CO_2_. When those gas compounds were expelled from the interior of the reduced GO film, micro-gasbags sprouted in assembled graphene nanosheets and contributed to the porous microstructures of GAF. The HRTEM image of GO nanosheets in Figure 2b demonstrates the wrinkled structure, which is beneficial for improved flexibility. On the other hand, the HRTEM image of GAF in Figure 1e shows the regular lattice fringe of the reduced GO nanosheets and declares it was highly graphitized. The cross-sectional SEM photo of GO and GAF are shown in Figure 1c,f for comparison. It can be seen that, after high-temperature reduction, the film expanded remarkably in thickness, and the dense microstructure of GO transformed into a porous foam-like structure. The 3D reconstruction of the morphology displays the rippled surface of GAF (Figure 2d) compared to the flat surface of GO (Figure 1g), confirming the porous microstructure of GAF.

To characterize the molecular structures of the films before and after reduction systematically, Raman spectroscopy was applied. As presented in Figure 2a, an obvious D peak at 1350 cm^−1^ could be observed, which informs the defective carbon structure in GO film. After annealing at 3000 °C, the D peak can hardly be observed for GAF, and instead, the intensity of the G peak at 1600 cm^−1^ increases significantly, which indicates its higher crystallinity than GO film. In addition, the new-emerged 2D peak at 2700 cm^−1^ confirms the existence of graphene layers. The ratio of *I*_D_/*I*_G_ was estimated to be 0.052, indicating the excellent crystal structure. The X-ray photoelectron spectroscopy (XPS) results further corroborate the much higher graphitization degree in GAF than in GO with diminished C-O peak and higher intensity of C-C peak at 284.5 eV (Figure 2b). The X-ray diffraction (XRD) peak at 2*θ* = 26.58° and 2*θ* = 10.95° (Figure 2c) indicates an interlayer spacing of *d* = 0.335 nm and *d* = 0.808 nm for GAF and GO, respectively. The decreased interlayer distance in GAF claims that numerous oxygen-containing functional groups in GO have been expelled by reduction. Fourier transform infrared spectroscopy (FTIR) measurement was also conducted to characterize the functional groups as depicted in Figure 2d. In contrast to GO, where several absorption peaks ascribing to -OH, -COOH, C-O-C, and -C=O were observed, no obvious absorption peaks were identified for GAF, confirming its high degree of reduction. Briefly, all these results verified the complete reduction of GO nanosheets to GAFs.

Electrical conductivity is one of the most important indicators to evaluate the ability to conduct electricity. To a specific specimen, the electrical conductivity σ can be calculated by *σ* = *Gl/A* where *G* is the conductance which is the reciprocal of the resistance *R*, *l*, and *A* are the length and cross-sectional area of the specimen, respectively. However, the result of traditional methods to measure the resistance between any two gauging points can hardly expose the conductance of specimens in practice. To understand the rules about the electrical conductivity of the obtained GAFs, the sheet resistance which is only related to the thickness of the given material has been used. A four-point probe is a commonly used equipment to measure the sheet resistance for conductive materials, especially films with relatively large lateral dimensions. Once sheet resistance *R_s_* is obtained, the conductivity can be determined using *σ* = 1/*R_s_d*, where *d* is the thickness of the specimen. This formula indicates that the most effective way to improve the electrical conductivity is by reducing the sheet resistance and thickness of the specimen. Given that the dense structure provides a superior electric conductive network, compressing the specimen with flat plates might be a feasible method. Thus, we treated the graphene films under different flat compression loads by the universal test machine and measured the conductivity for each c-GAF to systematically investigate the relation between applied stress and electrical conductivity of c-GAF.

The results of the thickness and electrical conductivity of c-GAFs under various stress are displayed in Figure 3a. From the figure, we can learn that with the applied compressive stress ranging from 0 to 450 MPa, the conductivity of c-GAF increased with increasing stress to a peak value of 5.37 × 10^5^ S/m and then showed a decrease to 4.36 × 10^5^ S/m with the stress of 550 MPa. Assume the symbol *p* as the compressive stress to describe the relationship between the conductivity and the applied pressure; the fit result is shown in Figure 3a as *σ* = −7 × 10^−10^*p*^4^ + 8 × 10^−7^*p*^3^ − 0.0003*p*^2^ + 0.0436*p* + 1.4289. Together with the monotonically increasing sheet resistance (Table 1), which is expected [33], we can conclude that the decreasing thickness upon increasing pressure dominated the improved electrical conductivity. A reasonable explanation for the increasing sheet resistance is the possible damage made to the graphene sheets during compression that finally destroys the initial conductive network of the assembled films. To gain an in-depth understanding of the relations between compression and conductivity, c-GAFs with 0 MPa, 150 MPa, 250 MPa, 350 MPa, 450 MPa, and 550 MPa (denoted as c-GAF-0, c-GAF-150, c-GAF-250, c-GAF-350, c-GAF-450, and c-GAF-550) were chosen for further measurements.

Figure 3b exhibits the top-view and cross-sectional (inset) optical images of those six representative samples. A wrinkled surface pitted with bulges can be observed for the c-GAF-0 sample, which may be due to the generation of the gaseous carbon oxides during the annealing process. With the increasing applied load, the surface first became smoother gradually till the pressure reached 350 MPa. Further increasing the compressive stress to 450 MPa, the lateral friction began to break the interfaces among the graphene layers. Giving the number of charge carriers is almost constant during this process, the number of the charge carriers per unit volume increased significantly, which is beneficial to the increase of the conductivity. The intensified damage finally accounted for the discontinuous surface on c-GAF-550. The cross-sectional images of the c-GAFs depicted in the insets of Figure 3b show decreasing thickness from 55 μm to 7 μm with increasing pressure. It is worth noting that the decrease is not linear due to the slowing expulsion rate of air bubbles with increasing compressive stress and decreasing thickness. Eventually, the thickness kept constant at 7 μm while the pressure exceeded 450 MPa.

The changes in microstructure in c-GAF could also be verified by the mechanical behavior in the tensile test for each specimen. Revealed by the load-strain curve and stress-strain curve in Figure 3c, the c-GAF-0 without compression manifests the best ultimate strain of about 2.55%, which benefits from its porous microstructure, and during the tension process, Young’s modulus kept changing when the strain exceeds about 0.75%. And the maximum section size of the c-GAF-0 accounts for its minimum strength (10.80 MPa) and Young’s modulus (1.04 GPa). Once the films were compressed to be dense, although the maximum loads were approximately equal, the ultimate strain decreased remarkably with increasing applied pressure. In contrast, the strength and Young’s modulus increased result of the decreasing thickness. Differently, the ultimate strain of the c-GAF-550 decreased dramatically compared with other compressed films and failed at a maximum load of only 1.21 N and an ultimate strain of 0.40%. As a typical laminated material, the comparison indicates that the continuity of the c-GAF-550 was most severely affected.

The result would be helpful in guiding the manufacture of GAF with excellent electrical conductivity. Encouraged by the result, we fabricated four strain sensors using c-GAF-0, c-GAF-150, c-GAF-450, and c-GAF-550 to further explore their applications. It is widely known that a strain sensor should be sensitive to convert the deformation of the object to an electric signal [34,38]. To analyze the relationship between the conductivity and sensitivity, specimens were cut into rectangular pieces, which are 5 mm wide and 80 mm long, with two copper electrodes fixed at each end by silver paint, as depicted in the inset of Figure 4a. Given that Young’s modulus of GAF is about 20 GPa, the c-GAF sensors were attached to the glass fiber reinforced plastic (GFRP) using PDMS to reduce the effect on measure result from adhesives.

In consideration of the mechanical properties of the c-GAF and GFRP, including elastic range and ultimate strain, the tensile tests for c-GAF strain sensors on GFRP were conducted by the test machine at a speed of 0.5 mm/min and maximum strain of 1%. During the stretching process, the resistance of different c-GAF strain sensors was collected at a frequency of 10 Hz by a data acquisition system, and the results are displayed in Figure 4b. As expected, the sensitivity of c-GAF-450 with the maximum conductivity was about 400% higher than c-GAF-0 with the minimum conductivity. It also demonstrates that the higher the conductivity of the sensor, the higher the sensitivity toward strain sensing, except for c-GAF-550. This could be attributed to the damaged inner structure of c-GAF-550; thus, with low strain, the conductive network failed to sense the deformation. When the strain was up to about 0.42%(failure point *f*), the damaged area could be observed, and the resistance increased dramatically. The sensor was broken completely when the strain approached about 0.60%; the resistance cannot be measured.

To verify the reliability of the c-GAF for application in strain sensor, the fatigue test with strain ranging from 0–0.6% (elastic range) was conducted with a c-GAF-450 sensor on the GFRP specimen. As depicted in Figure 4c, during the 200 cycles test, the resistance response of c-GAF-450 shows wonderful repeatability changing from 0% to 0.47% with the periodical strain. Two data pieces intercepted randomly from the curve were zoomed in for an examination, which showed almost identical patterns, indicating the superb stability and fast response of the c-GAF-450 sensor.

## 4. Conclusions

In summary, we proposed a universal strategy to tailor the electrical conductivity for the synthesized GAF by thermal annealing. The GAFs were compressed with stress ranging from 0 to 550 MPa to experimentally study the relation between the applied compress stress and the electrical conductivity. Encouragingly, by affecting both the thickness and the sheet resistance at the same time, the number of the charge carriers per unit volume was changed, and the electrical conductivity of the c-GAF was tunable ranged from 0.88 × 10^5^ S/m to a maximum value of 5.37 × 10^5^ S/m. It also proved that there is optimal stress at 450 MPa for c-GAF to gain the most excellent electrical conductivity, and further increasing the pressure resulted in decreasing conductivity. Furthermore, as part of the exploration of the application, comparative uniaxial tensile tests and fatigue tests for the performance of c-GAF sensors were conducted, which demonstrated the outstanding sensitivity and reliability of the c-GAF-450-based strain sensor. Based on the experimental results, our further work will focus on the conductive mechanism of the c-GAF and try to expose the rule of electrical conductivity theoretically.

## Figures and Tables

**Figure 1 materials-16-00526-f001:**
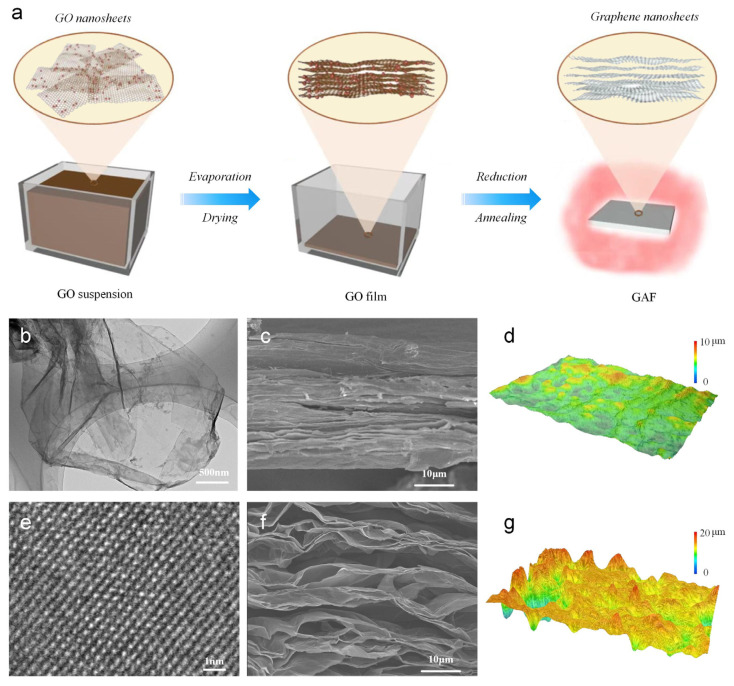
Reduction of the GO nanosheets. (**a**) Schematic illustration of the reduction process. (**b**) HRTEM image of GO nanosheets. (**c**) Cross-sectional SEM image of GO film. (**d**) The surface picture of the GAF taken by a super-large depth of field 3D microscopic system for GO film. (**e**) HRTEM image of rGO (**f**) SEM image of the obtained GAF. (**g**) The surface picture of the GAF taken by a super-large depth of field 3D microscopic system for GAF.

**Figure 2 materials-16-00526-f002:**
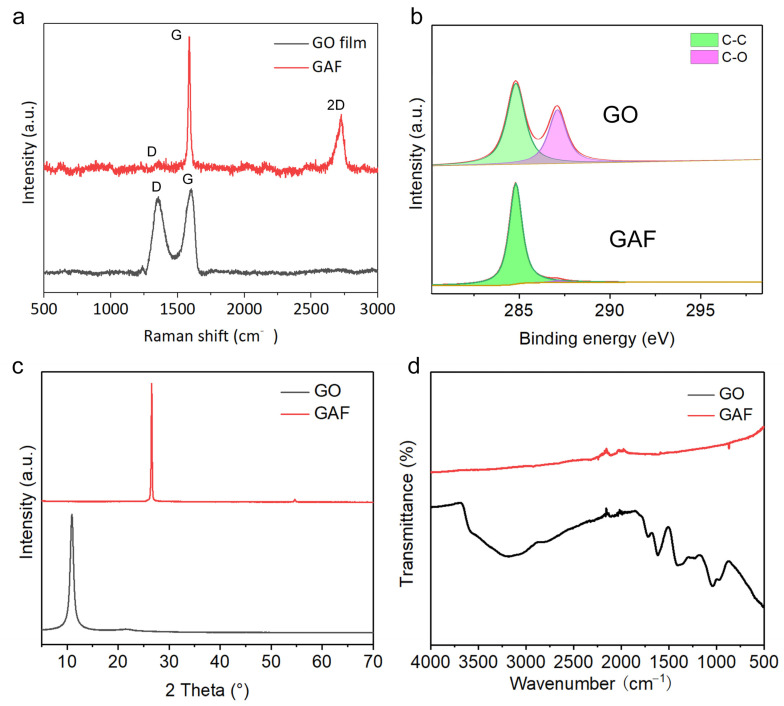
Characterizations of GO and GAF. (**a**) Raman spectra of GO and GAF. (**b**) XPS spectra of GO and GAF. (**c**) XRD patterns of GO and GAF. (**d**) FTIR spectra of GO and GAF.

**Figure 3 materials-16-00526-f003:**
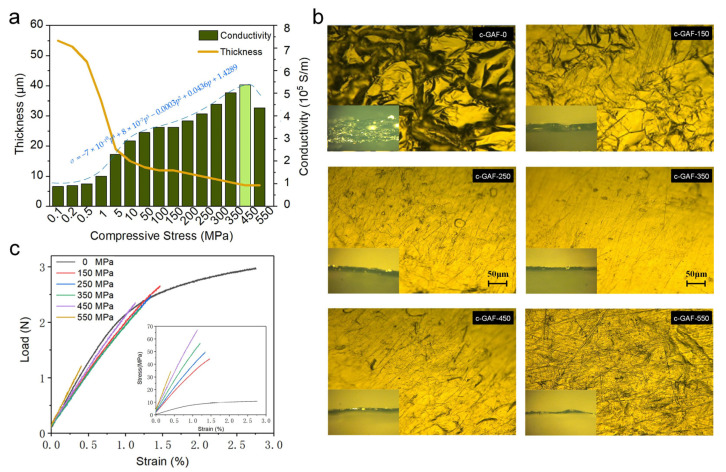
Measurement results of c-GAFs. (**a**) Electrical conductivity and thickness for c-GAFs. (**b**) Top-view and cross-sectional (inset) optical images of various c-GAFs. (**c**) Load-strain curve for c-GAFs.

**Figure 4 materials-16-00526-f004:**
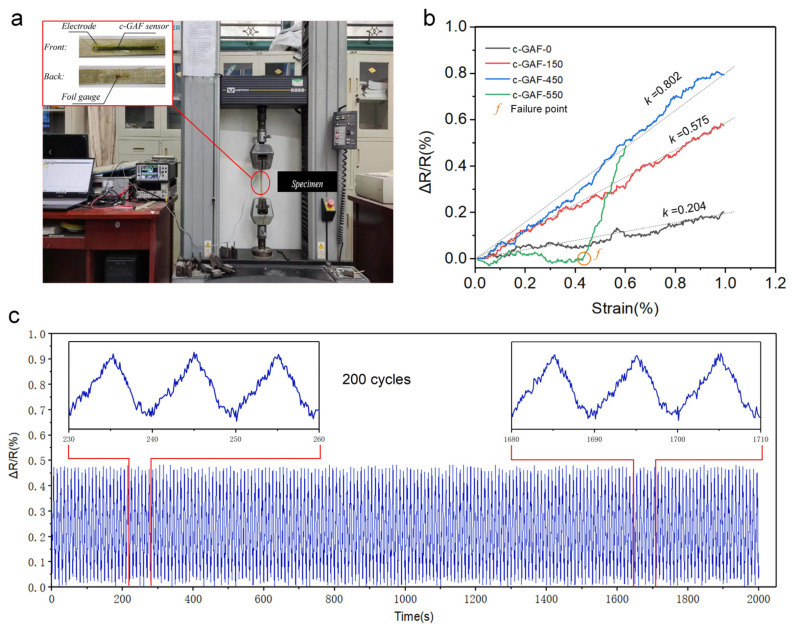
Performance of c-GAF strain sensors. (**a**) Photos of the specimen on the testing machine and the GFRP substrate with c-GAF and foil gauge strain sensor (inset). (**b**) Sensitivity test results for c-GAF strain sensors. (**c**) Resistance change of the c-GAF-450 strain sensor during the fatigue test ranged from 0–0.6%.

**Table 1 materials-16-00526-t001:** Summary of conductivity of c-GAF obtained with different pressures.

Compressive Stress (MPa)	Thickness (μm)	Sheet Resistance (mΩ/sq)	Conductivity (10^5^ S/m)
0.1	55	206	0.88
0.2	53	206	0.92
0.5	48	209	1.00
1	35	213	1.34
5	19	229	2.30
10	15	230	2.90
50	13	235	3.27
100	12	238	3.50
150	12	238	3.50
200	11	240	3.79
250	10	244	4.10
300	9	246	4.52
350	8	249	5.02
450	7	266	5.37
550	7	328	4.36

## Data Availability

Not applicable.

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
