# Peer review of "Compressed Graphene Assembled Film with Tunable Electrical Conductivity"

_materials, 2023, doi:10.3390/ma16020526_

Round 1
Reviewer 1 Report
This article is focused on improvement of the electrical conductivity of reduced graphene oxide (rGO) based graphene assembled films (GAFs) by applying series of compressive stress. The authors have systematically investigated the relation between the compressive stress and the electrical conductivity. It may may be published in Materials after major revision in the light of following comments:
1. The ratio of the intensity of D-band to G-band i.e. (ID/IG) should be mentioned which will clearly shows intact graphitic domain.
2. Young’s modulus and compressive strength percentage must be correlated to the applied strain.
3. Then a relation should also be established between conductivity and the preceding mechanical parameters since the application of c-GAF is intended as piezoelectric strain sensor.
4. According to thermodynamics, electrical conductivity is an intensive property, then how can it be related to sheet thickness? In my view, the sheet thickness (d) is a constant in the equation as it will be same for each sample while measuring the sheet resistance. If sheet thickness is not constant, then they are two different materials since in principle, the thickness and sheet resistance are unrelated. There is dire need of solid scientific reason to support you hypothesis.
5. Rather than thickness, it is the sample density which is affecting the sheet resistance as is apparent from your discussion. It means that the number of charge carriers per unit volume is increasing that is the charge density is increasing with increase in compressive stress. I will suggest that if possible, the charge density should be taken into consideration as well.
6. During the tensile tests which were also performed, does the sheet resistance increased or decreased with increase in elongation? Support you claim with the data that you have acquired using the data acquisition system. With sheet stretching, the thickness decreases which should increase the resistance according to you hypothesis.
7. There are few typos which should also be corrected such as “miscro-gasbags”.
Reviewer 2 Report
Dear Authors,
Your Manuscript, presents essentially an experimental investigation about the effect of pressure on the electric conductivity of a synthesized graphene assembled film (GAF). In addition, structural investigations by various techniques such as (SEM, HRTEM, Raman, XPS, WRD and FTIR) were also done and I wanted to congratulate you for this experimental achievement. However, if you have no connection within the text with theoretical work about electric conduction in graphene, please add some references about that. In addition, please check the following before resubmitting:
1) Please delete the back yellow coloring of certain text in your manuscript before resubmission.
2) In the abstract better to replace c-GAFs by compressed GAFs (c-GAFs) then you can use this abbreviation later in the text.
3) Please in scientific writing we avoid adverbs like “wonderful, beautiful, etc…” so please delete in line 45 wonderful and use simply these “works inspired so many…” the reader will understand the importance of these works.
4) Page 2, line 47-48 please delete this vulgarization sentence “Electrical conductivity is an extremely important fundamental property which represents the ability of the material to conduct current” you’re publishing in a specialized journal so everyone should know what electric conductivity means. Start directly by “electric conductivity is one of the determining properties ….”
5) In page 2 line 51 please delete “impressive”
6) Please replace “superb” in line 66 by “good”.
7) Please delete “universal” in line 245.
8) Could you justify to the reader why are you compressing GO between two PET films? What is the reason for choosing PET? why the compression is not done just directly?
9) Do you have any explanation for the existence of this optimal stress at 450 MPa ? Also for the increasing conductivity with pressure ? if not please open the perspective at the end of your work for connection with theoretical work on conduction in such media.
Best Regards
Reviewer 3 Report
In the manuscript entitled "Compressed Graphene assembled film with tunable electrical conductivity", the authors tried to improve the electrical conductivity of reduced graphene oxide (rGO) based graphene assembled films (GAFs) by applying a series of compressive stress and systematically investigated the relationship between the compressive stress and the electrical conductivity. However, the aim is not precise, and also the methodology. The discussion in the results section needs to be more specific. The English Language can be improved. So I recommended that the paper can be accepted after minor revision.
Round 2
Reviewer 1 Report
The authors have uploaded revised version of their manuscript for further review. They have complied most of the earlier comments. It is recommended for for publications in Materials.
Reviewer 2 Report
Dear Authors,
The manuscript has been improved according to the previously submitted suggestions and the article in its present form could be published in materials journal.
Best Regards